# Schwannoma of the Upper Limb: Retrospective Study of a Rare Tumor with Uncommon Locations

**DOI:** 10.3390/diagnostics12061319

**Published:** 2022-05-26

**Authors:** Mihaela Pertea, Alexandru Filip, Bogdan Huzum, Sorinel Lunca, Claudiu Carp, Mihaela Mitrea, Paula Toader, Stefana Luca, Dan Cristian Moraru, Vladimir Poroch, Bogdan Veliceasa

**Affiliations:** 1Faculty of Medicine, “Grigore T Popa” University of Medicine and Pharmacy, 700115 Iasi, Romania; alexandru-filip@umfiasi.ro (A.F.); bogdan.huzum93@gmail.com (B.H.); sorinel.lunca@umfiasi.ro (S.L.); adrianclaudiucarp@gmail.com (C.C.); mitrea.mihaela@umfiasi.ro (M.M.); stefana.luca@d.umfiasi.ro (S.L.); morarudc@gmail.com (D.C.M.); vladimir.poroch@umfiasi.ro (V.P.); bogdan.veliceasa@umfiasi.ro (B.V.); 2Department of Plastic Surgery and Reconstructive Microsurgery, “Sf. Spiridon” Emergency County Hospital, 700111 Iasi, Romania; 3Department of Orthopaedics and Traumatology, “Sf. Spiridon” Emergency County Hospital, 700111 Iasi, Romania; 4Second Surgery Clinic, Regional Institute of Oncology, 700483 Iasi, Romania; 5Department of Morphofunctional Sciences, “Grigore T Popa” University of Medicine and Pharmacy Iasi, 700115 Iasi, Romania; 6Department of Dermatology, CF Clinic Hospital, 700506 Iasi, Romania; 7Department of Palleative Care, Regional Institute of Oncology, 700483 Iasi, Romania

**Keywords:** schwannoma, upper limb, collateral digital nerves, enucleation, surgery

## Abstract

Background: Although schwannoma (neurilemmoma) is the most common tumor of the peripheral nerve, its low incidence, slow growth and vague symptoms often lead to misdiagnosis or delayed diagnosis. The aim of the study is to present a series with a large number of schwannomas in the upper limbs, some with very rare occurrence. Methods: We report 17 patients with a mean age of 58.5 years and upper limb schwannomas, located on the median, ulnar and radial nerves, but also on the posterior interosseous nerve and digital collateral nerves. The diagnosis was made by clinical examination and imaging tests, and in no case was a preoperative biopsy performed. Surgical treatment was established based on symptoms or aesthetic concerns. In all cases, a diagnosis of schwannoma was confirmed through histopathological and immunohistochemical examinations. Results: For all patients, a complete tumor enucleation was performed under a surgical microscope. No recurrence was recorded at 2 years after surgery. Patient satisfaction was good, with complete socio-professional integration in all cases. Conclusions: Although more frequently present on the main nerve trunks, schwannoma may be present on the collateral digital nerves in rare cases. A correct technique with complete tumor excision offers excellent postoperative outcomes and avoids recurrences.

## 1. Introduction

Schwannoma is the most common tumor of the peripheral nerve (approximately 90% of all nerve tumors) of ectodermal origin [1]. Frequently occurring in the mixed nerves, it accounts for approximately 5% of all soft tissue tumors [1,2]. Its distribution in anatomical regions is very different, being frequently found in the ear, nose, and throat regions, followed by the trunk and then by the upper (19%) and lower limbs (17.5%) [3]. It is more common between 30 and 60 years of age [4,5,6,7,8]. Due to the rarity of the tumor, its slow growth and the absence of specific symptoms, diagnosis is often difficult, late or incorrect [2,9,10,11]. This type of benign tumor of the peripheral nerve is most often located in the large nerve trunks, and in the upper limb is more often located on the anterior aspect, frequently involving the ulnar nerve, in over 75% of cases occurring distal to the elbow [2,12]. In 90% of cases, the schwannoma is a tumor with a single location [13]. The presence of multiple schwannomas is related to neurofibromatosis type 2 (NF2) [2,12]. Its clinical diagnosis is not based on specific signs, although the presence of a slow-growing tumor mass along the course of a nerve, that is mobile in the transverse plane, relatively fixed in the longitudinal plane, and sometimes painful on palpation may suggest a clinical diagnosis of schwannoma [12,14]. Imaging investigations—ultrasonography (USG) and, first of all, magnetic resonance imaging (MRI) description—have a major role in preoperative diagnosis, being considered helpful for establishing a correct diagnosis in 90% of cases [2,15,16,17]. None of these imaging investigations is 100% accurate in diagnosing schwannoma [18]. Surgical treatment is curative and ideally consists of enucleation of the tumor. Preoperative biopsy is not indicated when the presence of a schwannoma is suspected to avoid damage to the nerve bundles [15,19,20]. In the vast majority of cases, postoperative outcomes are positive with the preservation of intact peripheral nerve functions [21]. Criteria that may lead to an unfavorable postoperative outcome, some of which are in fact clinical signs of a high degree of initial nerve damage, have been described [22]. Definite diagnosis is obtained by a histopathological examination completed with immunohistochemical markers [23]. Recurrence or malignant transformation are very rare in schwannomas. The aim of this retrospective observational study is to conduct an analysis of the clinical and pathological findings and postoperative outcomes of a group of 17 patients (a large group), some with a very rare localization in the digital collateral nerves.

## 2. Materials and Methods

We studied a group of 17 patients with an age range of 31 to 85 years. The inclusion criterion was the histopathological diagnosis of schwannoma, and the location was only at the level of the upper limb. Patients in the study group were admitted to and treated at the Plastic Surgery and Reconstructive Microsurgery Clinic of the Iasi County “Sf Spiridon” Emergency Hospital, Romania between January 2014 and December 2018. Patients were informed about the therapeutic protocol, and all of them signed the informed consent. The current study was conducted with the approval of the Hospital Ethics Committee. The study group included 15 women and 2 men. The study method was retrospective and observational. We studied the following: patient age and sex, affected upper limb, affected nerve, tumor location, symptoms (pain, paresthesia, decreased muscle strength), Tinel sign, tumor size, preoperative imaging investigations, time of surgery, selected technique and the feasibility of performing surgery, the immediate outcomes, with the assessment of neurological symptoms (British Medical Research Council (BMRC), scale modified by Omer and Dellon), the result of the histopathological and immunohistochemical examinations, and the degree of patient satisfaction, which was assessed using Michigan Hand Outcomes Questionnaire (MHQ) scale. Imaging investigations were USG and/or MRI. Surgeries were performed under wide awake local anesthesia with the no tourniquet (Walant) technique when the tumor was located in the digital collateral nerves and under loco-regional anesthesia (axillary block) when the schwannoma affected the large nerve trunks located in the forearm and arm. For nerve tumors with axillary location, surgery was performed under general orotracheal intubation (OTI) anesthesia. When the extent of nerve damage allowed it, the tourniquet was used (except for the case in which the Walant technique was used) to create a bloodless surgical field and to improve visibility for complete tumor ablation and preservation of all nerve fibers. In all cases, schwannoma enucleation was performed under an operating microscope with microsurgical instruments. The assessment of outcomes was performed by sensitive and motor (active and passive) evaluation and confirmed by specific tests, such as two-point discrimination (2PD)and Semmes–Weinstein (SW) test. The postoperative follow-up period was 2 years after surgery. Patient satisfaction was assessed using MHQ. This questionnaire assesses how the patient feels and how well they can perform their usual activities.

## 3. Results

The study group included a total of 17 patients, of which 15 were female (88.2%) and 2 were male (11.7%), aged 31 to 85 years. Age distribution showed a total of five patients (29.4%) in the 30–50 age range, six patients (35.2%) in the 50–70 age range, and six patients (35.2%) over 70 years, with a mean age of 58.5 years (Table 1).

Tumors were located as follows: one case (5.8%) in the axilla, three cases (17.6%) at the arm level, seven cases (41.1%) at the forearm level, one case (5.8%) at the level of the wrist, and five cases (29.4%) in the hand (palm and fingers) (Table 2).

In our study group the following nerves were involved: ulnar nerve, seven cases (41.1%); median nerve, five cases (29.4%); radial nerve, one case (5.8%); digital nerves, three cases (17.6%); and posterior interosseous nerve, one case (5.8%) (Figure 1, Figure 2 and Figure 3). Of the 17 patients, in only two cases, schwannoma occurred on the dorsal aspect of the upper limb (radial nerve and posterior interosseous nerve).

Upon clinical evaluation, a tumor mass was detected, relatively well-defined, mobile in the transverse direction and fixed when mobilized in the axis of the limb. In most cases (88.2%), the clinical symptoms were represented by the presence of paresthesia and pain of varying intensity. No other neurological signs were described. In two of the 17 cases (11.8%), no clinical signs were recorded, the only complaint being an unsightly appearance due to the tumor size (Table 3).

The Tinel sign at the level of the tumor mass was positive in all cases. An ultrasound examination was performed in all cases, and MRI was performed in 11 patients (64.7%). The magnetic resonance imaging showed a tumoral mass with a moderate hyperintense feature in T1-weighted images and hyperintense feature on fluid sensitive sequences (Figure 4).

Surgery was indicated in all cases. It was performed under general OTI anesthesia in one case (axillary schwannoma), infraclavicular block in thirteen cases (arm, forearm, wrist and palm), and the Walant technique in three cases (fingers). For the Walant technique, 1% lidocaine and 1:100,000 epinephrine was used. No anesthesia-related immediate or late incidents or accidents were reported. In all cases, the surgical technique consisted of the enucleation of the tumor (schwannoma). An operating microscope and microsurgical instruments were used in all cases (Figure 5). 

Tumor sizes ranged from 0.7/0.4/0.4 cm to 7.5/5.5/4.5 cm (Table 3). The exeresis specimen was sent for histopathological examination. In all cases, a definite diagnosis of schwannoma was made by the presence of the characteristic appearance: compact hypercellular (Antoni A) and myxoid hypocellular areas (Antoni B) (Figure 6A,B). Additionally, nuclear palisades were described around the fibrillary processes (Verocay bodies) (Figure 6C).

In two of the case studies, cystic degeneration was detected. Immunohistochemical staining was performed: S100 (12 of the 17 cases) was positive, confirming the diagnosis of schwannoma and excluding the malignant tumors of the peripheral nerve sheath from the differential diagnosis, in which S100 was weak or negative (Figure 7).

Other immunohistochemical stainings were also used: CD34, collagen IV and glial fibrillary acidic protein (GFAP) (Figure 8, Figure 9 and Figure 10).

Postoperatively, none of the patients reported neurological signs and symptoms. In all cases, motor and sensitive functions were assessed in the immediate postoperative period, as well as at 2 years. The MHQ scale showed full satisfaction with socio-professional integration for all 17 patients.

## 4. Discussion

Herein, we present a large series of 17 patients with upper limb schwannoma, emphasizing the rare occurrence of this tumor in the hand. Schwannoma, also known as neurilemmoma, was first described by Verocay in 1980 as being the most common tumor of the peripheral nerve. Schwannoma accounts for about 5% of all soft-tissue tumors [24]. Its etiology is not well-known and appears to have chromosomal abnormalities with the involvement of chromosome 22 [24]. Over time, schwannoma was reported to be more common between 30 and 60 years of age, with no preference regarding gender [25]. Most often, schwannoma is a solitary tumor, occurring sporadically (90% of cases), slow-growing, and with few symptoms [2,13,22]. The incidence of cases with multiple schwannomas is reported to range between 9% and 13% [23,24,25]. Schwannoma may be associated with neurofibromatosis type 2 (NF2) in 3% of cases, schwannomatosis in 2% of cases, and or meningiomatosis associated or not associated with NF2 [26]. The malignant transformation of this type of nerve tumor is extremely rare [20,26]. Regarding its location in the upper limb, it seems that ulnar nerve involvement is the most common, the anterior aspect of this anatomical segment being the most often affected [2,3,14]. An interesting finding in our study was that the mean age of the patients was close to 60 years, higher than in other reported studies. Adani et al. reported a mean age of 44, Majumder et al. reported 48 years, and Hirai and Tang reported 56 years [2,6,22,27]. Of the 17 patients, only 2 were male, with a female:male ratio of 15:2 compared to an approximate 1:1 ratio in other studies [27]. As reported in the literature, the most frequently involved nerve in our study patients was the ulnar nerve (in 7 of the 17 cases). Additionally, we reported three cases of schwannoma in the digital collateral nerves, a very rare finding [28]. Regarding the history of the disease, it extended over a period of between 2 and 5 years. Cases with a longer disease course were reported in the literature [29]. Being a rare tumor, the schwannoma can often be misdiagnosed and “confused” with lipoma, neurofibroma, fibroma, ganglion, and xanthoma [10,30]. In 14 of the 17 cases of our study, the tumor was identified on the anterior aspect of the anatomical segment of the upper limb, this being in agreement with the data in the literature [2]. Clinical findings may be suggestive of the diagnosis when neurological symptoms are present. There are no pathognomonic clinical signs to diagnose schwannoma and no clinical signs to clinically differentiate between a schwannoma and neurofibroma [1]. In our study, patients reported minimal pain as well as mild paresthesia, and in all cases, the Tinel sign was positive. The positive Tinel sign (with a high predictive value of 87.5%) is associated with imaging tests and caused a fairly high concordance between the presumptive and the definitive diagnosis (over 90%) [2]. The Tinel sign is present in 4% to 76% of cases, as reported by Artico et al. in 1996 [31]. As reported by Gosk et al., the presence of major neurological signs initially represents an unfavorable prognostic factor in terms of postoperative results [15]. USG and MRI imaging tests play an important role in guiding preoperative diagnoses, since they do not have 100% accuracy [32,33]. On USG, schwannomas are usually seen as homogeneous, well-defined hypoechoic masses, often ovoid, and can show the origin and relationship of the tumor with the affected nerve [27]. MRI remains the most useful investigation with a diagnostic value of over 90%. Schwannoma, like other peripheral nerve tumors, is seen on MRI as moderately bright in proton density-weighted images and bright in T2-weighted images [33]. In our study group, USG was performed in all cases, while MRI was performed in only 10 of the 17 cases, sometimes due to financial reasons. The treatment of schwannoma is surgical and consists of tumor enucleation under a surgical microscope [15,18,22,33,34]. In our study, this technique could be used in all cases so that the risk of nerve damage was minimal. This was reflected in the postoperative outcomes showing no neurological impairments, or worse compared to the preoperative data. Rodrigues et al. reported a schwannoma of the ulnar nerve of 7.5/3.5/2.7 cm located on the distal forearm [1]. In our study group, the size of the largest schwannoma was 7.5/5.5/4.5 cm, and the average size was 2.5/1.9/1.6 cm. Biopsy in case of a presumptive diagnosis of schwannoma is not indicated to avoid the iatrogenic injuries to nerve fibers [2,15,18]. The definite diagnosis is made by pathological and immunohistochemical examinations. The microscopic appearance of schwannomas is characterized by the presence of two distinct areas: compact hypercellular (Antoni A) and myxoid hypocellular (Antoni B), as well as nuclear palisades around the fibrillary processes (Verocay bodies), present in cellular areas [25,27]. In two of the 17 cases, cystic degeneration was recorded. Hirai et al., in their 2019 study of 139 patients with schwannoma, reported the presence of cyst formation, fibrosis, and calcification after bleeding upon a microscopic examination of degenerative changes [22]. Immunohistochemical staining was positive in all cases for S100, confirming the diagnosis and differentiating it from neurofibroma. In some cases, immunostaining for GFAP was done, and it was positive, while SMA was negative. Intraoperative and postoperative complications were not reported in any of our 17 cases. Knight, Birch and Pringle, in their study of 198 patients recorded complications in only 5 patients, including those who underwent nerve graft reconstruction [11]. No neurological signs and negative prognostic factors were reported in any of our patients. Because schwannoma is a rare tumor with nonspecific symptoms, the surgeon should include it in the differential diagnosis of soft tissue tumors [28,30,35]. Our study had an inclusion criterion of a histopathological diagnosis of schwannoma; therefore, we did not assess the degree of concordance between the presumptive and definitive diagnosis. The outcomes are positive and without neurological complications when the surgical technique is precise, the nerves are not traumatized, the schwannoma is completely removed, and the nerve fibers are kept intact. The recurrence of schwannoma is rarely reported and is directly related to its incomplete excision [24]. In the current study, no recurrence was recorded during the 2-year follow-up.

## 5. Conclusions

Schwannoma, a rare benign tumor, must be recognized in order to avoid a misdiagnosis or late diagnosis that may result in significant neurological damage. Although reported to be most common during the third to the sixth decades of life, it may also occur later, even in the ninth decade of life. Although the symptoms are nonspecific or even absent, sometimes, the association of a positive Tinel sign and imaging tests (USG and MRI) may suggest a diagnosis of schwannoma. Although more frequently present in the main nerve trunks, schwannoma may be rarely present on the collateral digital nerves. If a peripheral nerve tumor is suspected, a preoperative biopsy should not be performed to avoid iatrogenic lesions. If the tumor proves to be malignant (intraoperatively or postoperatively), an oncological R0 is necessary. The treatment is surgical, consisting of tumor enucleation using microsurgical techniques and instruments to avoid additional nerve trauma and preserve unaffected nerve fibers. A clean technique with complete tumor excision is the solution for excellent postoperative outcomes and the avoidance of recurrences.

## Figures and Tables

**Figure 1 diagnostics-12-01319-f001:**
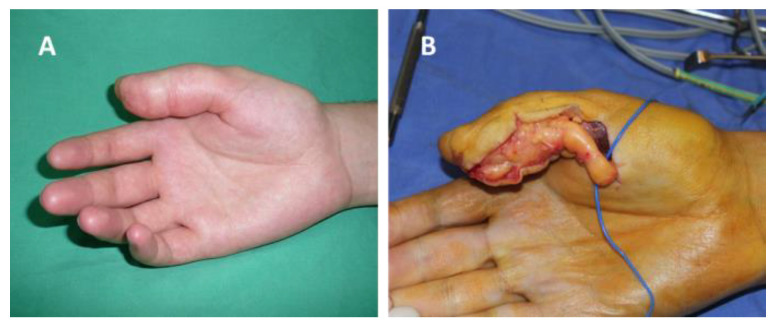
Schwannoma of the radial collateral nerve of the right thumb: (**A**) before surgery, (**B**) intraoperative aspect.

**Figure 2 diagnostics-12-01319-f002:**
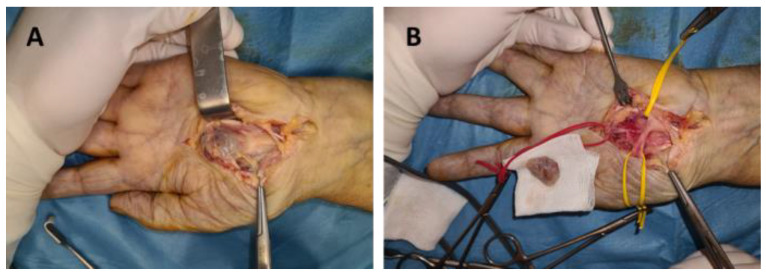
Schwannoma of the median nerve in the palm: (**A**) dissection of the tumor, (**B**) tumor enucleation.

**Figure 3 diagnostics-12-01319-f003:**
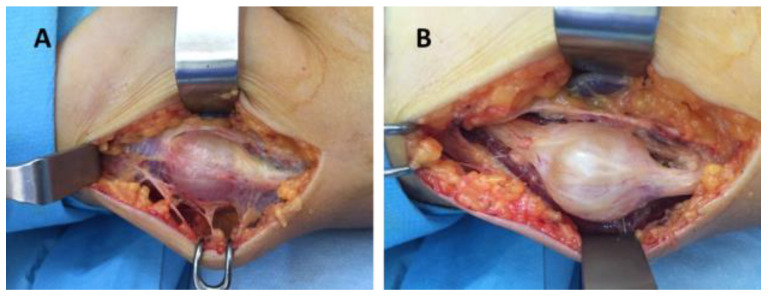
Schwannoma of the radial nerve: (**A**,**B**) intraoperative aspects.

**Figure 4 diagnostics-12-01319-f004:**
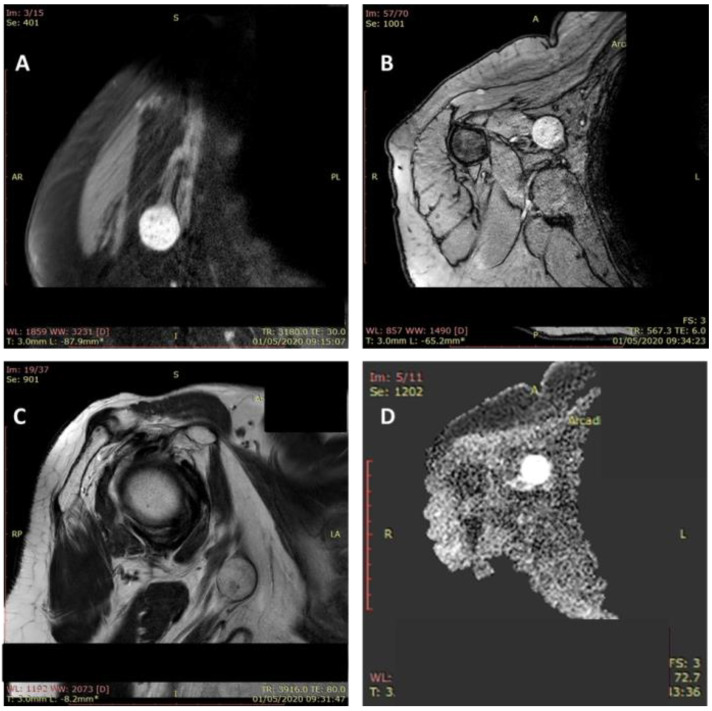
MRI aspect of a schwannoma of the median nerve: axillary localization. Tumoral mass (2.5/2/1.1 cm) with a moderate hyperintense feature in T1-weighted images and hyperintense feature on fluid sensitive sequences (**A**–**D**).

**Figure 5 diagnostics-12-01319-f005:**
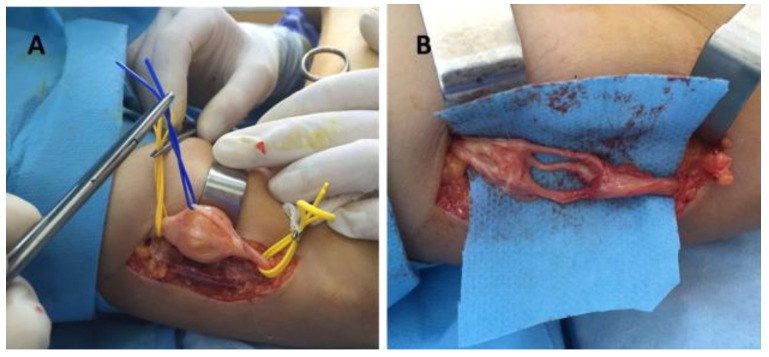
Schwannoma of the radial nerve—intraoperative aspect: (**A**) tumor dissection and isolation of the unaffected fibers, (**B**) radial nerve after schwannoma enucleation.

**Figure 6 diagnostics-12-01319-f006:**
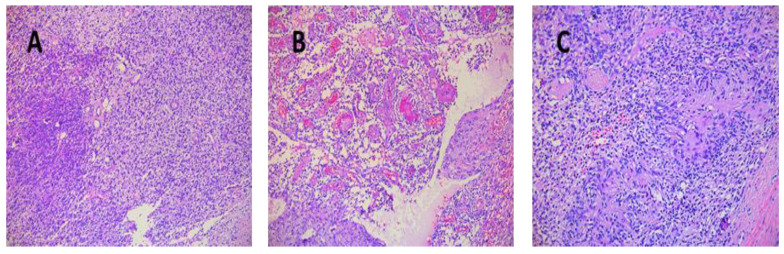
(**A**) Schwannoma—dense zone, loose zone HE × 10. (**B**) Schwannoma—vessels with hyaline walls HE × 10. (**C**) Schwannoma—Verocay bodies HE × 10.

**Figure 7 diagnostics-12-01319-f007:**
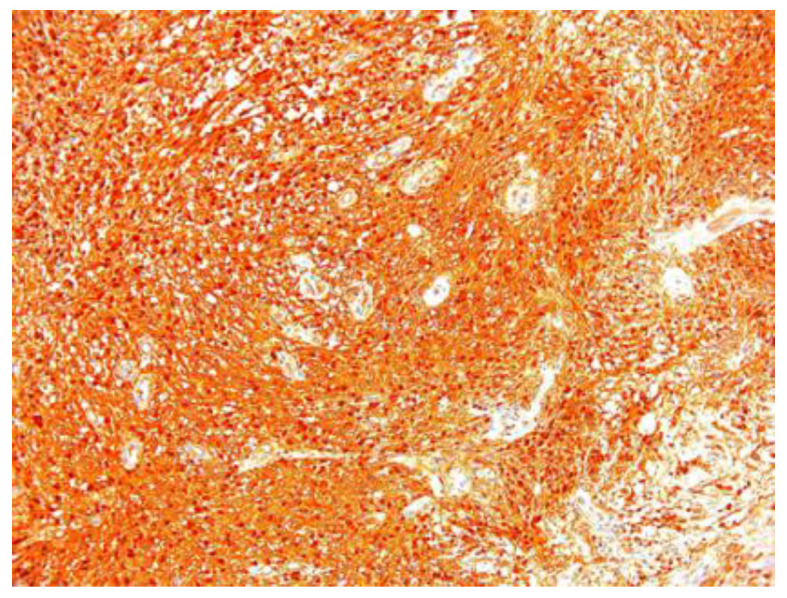
Schwannoma S100 positive ×10.

**Figure 8 diagnostics-12-01319-f008:**
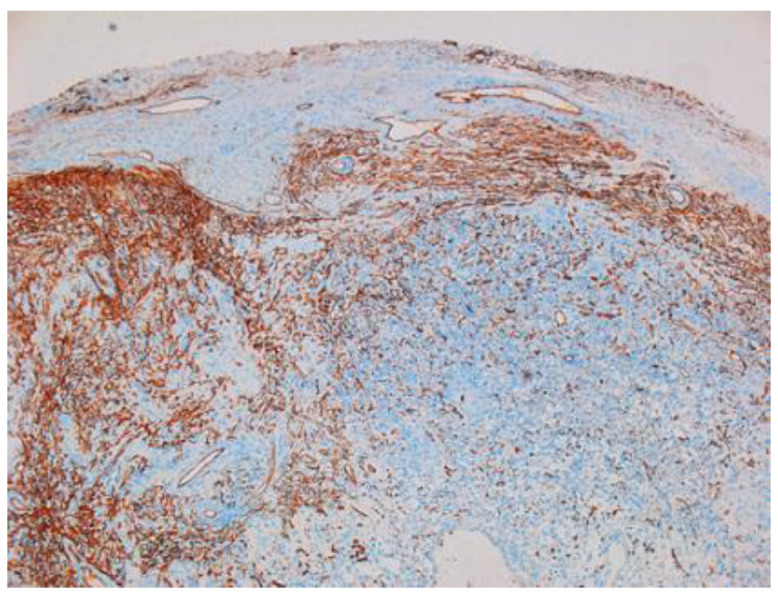
Schwannoma—CD34-positive subcapsular marking ×2.5.

**Figure 9 diagnostics-12-01319-f009:**
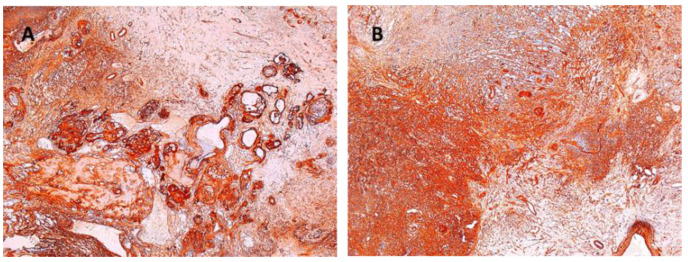
(**A**) Collagen IV positive in vascular wall of the schwannoma ×5. (**B**) Collagen IV positive in schwannoma ×5.

**Figure 10 diagnostics-12-01319-f010:**
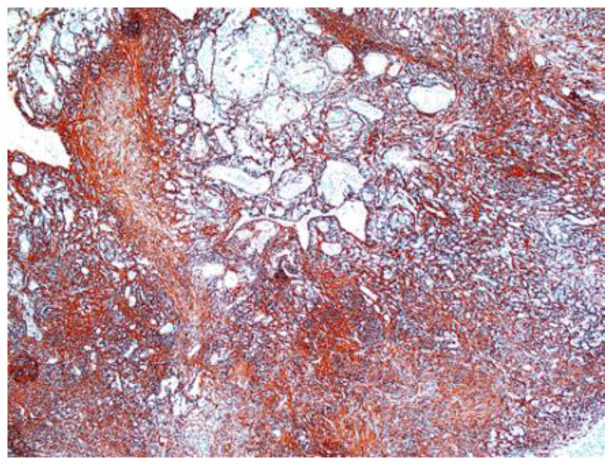
GFAP-positive in schwannoma ×5.

**Table 1 diagnostics-12-01319-t001:** Epidemiological and clinical features of the study group.

	Age	Sex	Axilla/ Arm	Elbow Wrist Hand	Anatomical Localization	Affected Thoracic Limb	Nerve Affected	Clinical Symptoms	USG/ MRI
1	71	F	-	+	forearm	L	ulnar	paresthesia minimal pain Tinel+	USG
2	53	F	-	+	forearm	L	ulnar	paresthesia Tinel+	USG MRI
3	85	F	+	-	axilla	R	median	paresthesia pain Tinel+	USG MRI
4	60	F	-	+	forearm	R	ulnar	paresthesia Tinel+	USG
5	48	F	+	-	arm	L	ulnar	paresthesia Tinel+	USG MRI
6	36	F	-	+	forearm	L	posterior interosseous	paresthesia Tinel+	USG MRI
7	52	F	-	+	forearm	L	ulnar	paresthesia medium pain Tinel+	USG MRI
8	77	M	-	+	palm	R	median	paresthesia Tinel+	USG MRI
9	72	F	-	+	palmar face of the second phalanx of the ring finger	L	collateral digital	paresthesia Tinel+	USG
10	57	F	+	-	arm	R	median	paresthesia Tinel+	USG
11	74	M	-	+	palmar face of the first phalanx of the little finger	L	collateral digital	paresthesia hipoesthesia Tinel+	USG
12	56	F	-	+	palmar face of the first phalanx of the thumb	R	collateral digital	paresthesia Tinel+	USG
13	38	F	-	+	anterior face of the wrist	R	median	paresthesia Tinel+	USG MRI
14	75	F	-	+	palm	L	median	paresthesia Tinel+	USG MRI
15	31	F	-	+	forearm	L	ulnar	Tinel+	USG MRI
16	62	F	-	+	forearm	R	ulnar	Tinel+	USG MRI
17	48	F	+	-	arm	R	radial	paresthesia Tinel+	USG MRI

M = male, F = female, R = right, L = left, USG = ultrasonography, MRI = magnetic resonance imaging.

**Table 2 diagnostics-12-01319-t002:** Affected nerves and the level of the tumor in the study group.

	Affected Nerve	Cases Number	Level of Nerve Tumor
1	ulnar	7	Arm—1 Forearm—6
2	median	5	Axilla—1 Arm—1 Wrist—1 palm 2
3	collateral digital	3	first phalanx of the little finger—1 first phalanx of the thumb—1 second phalanx of the ring finger—1
4	posterior interosseous	1	forearm
5	radial	1	arm

**Table 3 diagnostics-12-01319-t003:** Tumor sizes and immunohistochemical analysis of the study group.

	Age	Sex	Surgical Technique	Specimen Size (cm)	IHC
1	71	F	enucleation	2.4/2.3/1.5	−
2	53	F	enucleation	7.5/5.5/4.5	S100+, CD34+, SMA−
3	85	F	enucleation	2.5/2/1.1	S100+, CD34+, SMA−
4	60	F	enucleation	1.3/0.9/0.7	−
5	48	F	enucleation	2.4/1.8/1.5	S100+, collagen IV+, SMA−
6	36	F	enucleation	0.7/0.4/0.4	S100+
7	52	F	enucleation	3/2.7/2.5	S100+, collagen IV+, SMA−
8	77	M	enucleation	4.1/3.2/2.7	S100+, SMA−
9	72	F	enucleation	1.4/1.1/0.9	S100+
10	57	F	enucleation	3.1/2.8/2.6	S100+, SMA−,
11	74	M	enucleation	1.1/0.5/0.5	−
12	56	F	enucleation	2.2/2.1/1.3	S100+, collagen IV+, CD34+, SMA−
13	38	F	enucleation	1.2/0.5/0.7	−
14	75	F	enucleation	2.1/1.1/1.5	S100+
15	31	F	enucleation	3.2/1.9/1.8	S100+, SMA−
16	62	F	enucleation	3.1/1.8/1.4	S100+, CD34+, collagen IV+, SMA−
17	48	F	enucleation	2.4/1.8/1.5	−

M = male, F = female, cm = centimeters, IHC = immunohistochemical tests.

## Data Availability

Not applicable.

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
