# Peer review of "Schwannoma of the Upper Limb: Retrospective Study of a Rare Tumor with Uncommon Locations"

_diagnostics, 2022, doi:10.3390/diagnostics12061319_

Round 1
Reviewer 1 Report
I would recommend publishing - but there are some english edits necessary - for example page 2/11 line 69 "thoracic limb" is an unusual term and should be replaced by upper limb.
Line 91 makes reference to nerve "fillets" and I think that this should be fibres.
The last sentence in your conclusion should be clarified - you recommend no biopsy - but what if you suspect a malignant peripheral nerve sheath tumor? - then biopsy and pre-op RT might be indicated. You should specify that you are referring to benign nerve sheath tumors. Thanks
Author Response
Dear Reviewer
Thank you for your appreciations and recommendations.
According to your recommendations we made the corrections indicated on lines 69 and 91 and also other editing corrections in English.
I hope that I have made the necessary clarifications regarding the conclusions of the article, as you indicate.
Thank you very much,
Best regards,
Mihaela Pertea MD PhD,
author of the manuscript ID – diagnostics 1726208
Reviewer 2 Report
Major comments:
In the section of “introduction” the concepts are clear, but, from my point of view, the authors should improve the quality of exposure because it seems to be an article of scientific magazine, but not a scientific article. Please, revise!
Line 69: “thoracic limb” is a bad definition. Please, revise!
Line 75: “The following were studied”. What? Please, add subject
Line 84: “USG and /or MRI.”. Please, write whole word the first time
Line 87: “surgery was performed the surgery was performed”. Please, correct!
Line 91: “all nerve fillets”. This is a wrong exposure. Correct
Line 93-98: We’re not there yet. The sentences are write very bad. It not seems a scientific article.
Line 100: “Michigan Hand Outcomes Questionnaire (MHQ).” Please, explain shortly what it is.
Line 102-103: sorry but this is not admissible. Is not possible to submit a manuscript without read it before. I stop the peer-review e suggest to the authors to change and check the entire manuscript.
Author Response
Dear Reviewer
Thank you for your appreciations and recommendations.
According to your recommendations we made the corrections indicated in Introduction and on lines 69, 75, and 91 and also other editing corrections in English.
“USG” and “MRI” are mentioned for the first time in the article at line 51 and they are written with whole word.
I made the corrections you indicated on the lines 87 and 91 and I have clarified the text between lines 93 si 98 as you indicate.
At line 100 I made a short explanation of MHQ questionnaire.
Sorry for lines 102-103! I made the necessary corrections.
I also made a series of corrections in the text at “abstract”, “introduction”, “material and method”, “results”, “discussions” and “conclusions” as well as corrections on English editing.
Thank you very much,
Best regards,
Mihaela Pertea MD PhD
author of manuscript ID diagnostics-1726208
Reviewer 3 Report
An interesting review of this subject which merits publication.
Well written with useful relevant figures
Accept
Author Response
Dear Reviewer
Thank you for your appreciations.
Thank you very much
Best regards,
Mihaela Pertea MD PhD
author of manuscript ID diagnostics-1726208
Round 2
Reviewer 2 Report
The authors improved the quality of manuscript.